# One-year recovery rates for young people with depression and/or anxiety not receiving treatment: a systematic review and meta-analysis

Anna Roach ![ORCID],[1] Diliniya Stanislaus Sureshkumar,[1] Kathryn Elliot,[1] Liliana Hidalgo-Padilla,[2] Francois van Loggerenberg,[1,3] Lauren Hounsell,[1] Zivile Jakaite,[1] Fernando Esnal,[4] Jade Donaghy,[1] Victoria Jane Bird,[1] Stefan Priebe[1]

## ABSTRACT

**Objectives** To systematically review 1-year recovery rates for young people experiencing depression and/or anxiety who are not receiving any specific mental health treatment.

**Design** Systematic review and meta-analysis.

**Data sources** MEDLINE, Embase, PsycINFO, Web of Science and Global Health were searched for articles published from 1980 through to August 2022.

**Eligibility criteria** Articles were peer-reviewed, published in English and had baseline and 1-year follow-up depression and/or anxiety outcomes for young people aged 10–24 years without specific treatment.

**Data extraction and synthesis** Three reviewers extracted relevant data. Meta-analysis was conducted to calculate the proportion of individuals classified as recovered after 1 year. The quality of evidence was assessed by the Newcastle-Ottawa Scale.

**Results** Of the 17 250 references screened for inclusion, five articles with 1011 participants in total were included. Studies reported a 1-year recovery rate of between 47% and 64%. In the meta-analysis, the overall pooled proportion of recovered young people is 0.54 (0.45 to 0.63).

**Conclusions** The findings suggest that after 1 year about 54% of young people with symptoms of anxiety and/or depression recover without any specific mental health treatment. Future research should identify individual characteristics predicting recovery and explore resources and activities which may help young people recover from depression and/or anxiety.

**PROSPERO registration number** CRD42021251556.

## STRENGTHS AND LIMITATIONS OF THIS STUDY

⇒ The study design made it possible to accumulate and present research across multiple studies, countries and contexts.

⇒ Thorough literature search of five major electronic databases and reporting as per Preferred Reporting Items for Systematic Reviews and Meta-Analyses guidelines.

⇒ Evidence base with data of a total of 1011 participants is limited, and the small number of studies meant moderator analyses could not be conducted.

⇒ No available information about how long symptoms had been present for at baseline meaning participants with very short episodes of depression and/or anxiety were likely missed.

For numbered affiliations see end of article.

**Correspondence to**
Diliniya Stanislaus Sureshkumar;
d.s.sureshkumar@qmul.ac.uk

## INTRODUCTION

Adolescence and young adulthood are important developmental periods of social, behavioural and psychological change where there is a transition from the dependence of childhood to adulthood independence.[1] However, these periods are also associated with increased risk of experiencing mental distress with most functional mental disorders beginning before the age of 25 years.[2]

Research suggests that the majority of young people are likely to experience mental distress at some point of time,[3] yet the spontaneous prognosis of such distress remains uncertain. An understanding of the prognosis without treatment may inform appropriate responses on individual and public health levels.[4]

Improving mental health among young people has been identified by the WHO as a key priority to promote social and economic development.[5] Experiencing depression and anxiety in early life is associated with poor outcomes, such as high levels of distress and disability, future physical and psychiatric morbidity, and educational and social impairment.[6–8] Strengthening the resilience of young people is suggested as key to promoting long-term mental health outcomes.[9]

Although some research estimates that a proportion of those who experience depressive episodes in adolescence will go on to experience at least one recurrent episode in adulthood,[10] to date there has been no

systematic exploration of the 1-year recovery outcomes of adolescent depression and/or anxiety.

Against this background, we conducted a systematic review and meta-analysis of recovery rates of young people with depression and/or generalised anxiety who did not receive any specific mental health treatment. For an inclusive approach, we used a wide age range including adolescence and young adulthood, that is, 10–24 years, following the WHO definition of young people.[11] The 1-year time frame appears relevant for considering whether young people with symptoms of depression and/or anxiety should be referred to mental health services and has the pragmatic advantage that studies tend to report outcomes for 1 year rather than for other timeframes.

## METHODS

We conducted a systematic review and meta-analysis to estimate the proportion of young people who recover from depression and anxiety in a 1-year period. Methodology and reporting for this systematic review are consistent with Preferred Reporting Items for Systematic Reviews and Meta-Analyses (PRISMA) guidelines.[12] A PRISMA checklist is provided in online supplemental material 1. A protocol for this review was developed a priori and registered on PROSPERO and is available in online supplemental material 2.

### Search strategy

A comprehensive literature search was conducted in five electronic databases: MEDLINE, Embase, PsycINFO, Web of Science and Global Health for articles published from 1980 through to August 2022. The search strategy was adapted for the different databases and search terms, included the study design (longitudinal, cohort, prospective, etc), terms relating to young people (adolescents, youth, school, etc), terms encompassing mental distress (depression, anxiety, internalising disorder, etc) and a 1-year or 12-month follow-up. An example search strategy and the MeSH headings used are available in online supplemental material 3. The reference and citation lists were also scanned of eligible articles to supplement our database searches.

### Eligibility criteria

Articles were eligible if they included:
- ► Participants defined by WHO as 'young people' (aged 10–24) at baseline, which includes adolescents (aged 10–19) and youth (aged 15–24).
- ► A validated self-report or observer-rated measure of depression and/or generalised anxiety or a diagnosis of depression and/or generalised anxiety as defined by the ICD-10 or equivalent.
- ► A prospective design including 1-year follow-up depression and/or anxiety outcomes.

Only studies reporting individual recovery rates of anxiety and/or depression were included, studies reporting group recovery rates were not included unless

the whole sample had symptoms with anxiety and/or depression at baseline. Furthermore, included articles had to be published in English in peer-reviewed journals with longitudinal cohort research study designs (with 1-year follow-up data available).

We excluded all studies where young people were receiving pharmacological or psychological treatment or other specific interventions for their mental health. Grey literature was also not included.

### Data collection and analysis

A large team of independent reviewers (AR, DSS, LH-P, FvL, ZJ, LH, FE, JD) screened titles and abstracts to ascertain potential eligibility. Those appearing relevant underwent full-text review, of which 20% were double screened (KE). The disagreement rate was 3%, and the conflicts were resolved by an independent reviewer (FvL).

### Data extraction

Three reviewers (AR, DSS and KE) independently extracted data from eligible articles into a computerised extraction form developed before review (available on request). Data extracted included study objective, country/setting of research, sample size, baseline outcome measure, validated measure of depressive and/or anxiety symptoms (including if self-report/clinician led), age at baseline, follow-up duration, follow-up outcome, method of analysis and covariates.

### Risk of bias

The methodological quality of eligible articles was assessed using the Newcastle-Ottawa Scale (NOS)[13] a quality assessment tool used for non-randomised controlled trial (non-RCT) studies including case–control and cohort studies. The NOS uses a star rating system; stars are assigned for eight items grouped into three categories assessing selection of study groups, comparability and outcomes.

### Data analysis

A meta-analysis was conducted within STATA to calculate the proportion of individuals classified as recovered. We used Wilson's method[14] to calculate 95% CIs for proportion estimates. The approach produces asymmetric CIs in studies with low proportion rates. Heterogeneity among studies was estimated based on Cochran's Q and reported using $I^2$ (and 95% CI of the $I^2$). $I^2 > 75\%$ is considered indicative of high heterogeneity. Given the heterogeneity within the papers, we used a random effects meta-analysis applying the metan code within STATA V.15.[15]

As it is common for meta-analyses of observational or cohort data to report high levels of heterogeneity, prediction intervals (PIs) were also calculated. The prediction interval includes the plausible range of estimates we would expect the effect of a new study to be within. They are wider than CIs as they include the random variation of individual measurement as well as the uncertainty of estimating the population effect size. To calculate the PIs, the 'meta set' command was first used to declare the data

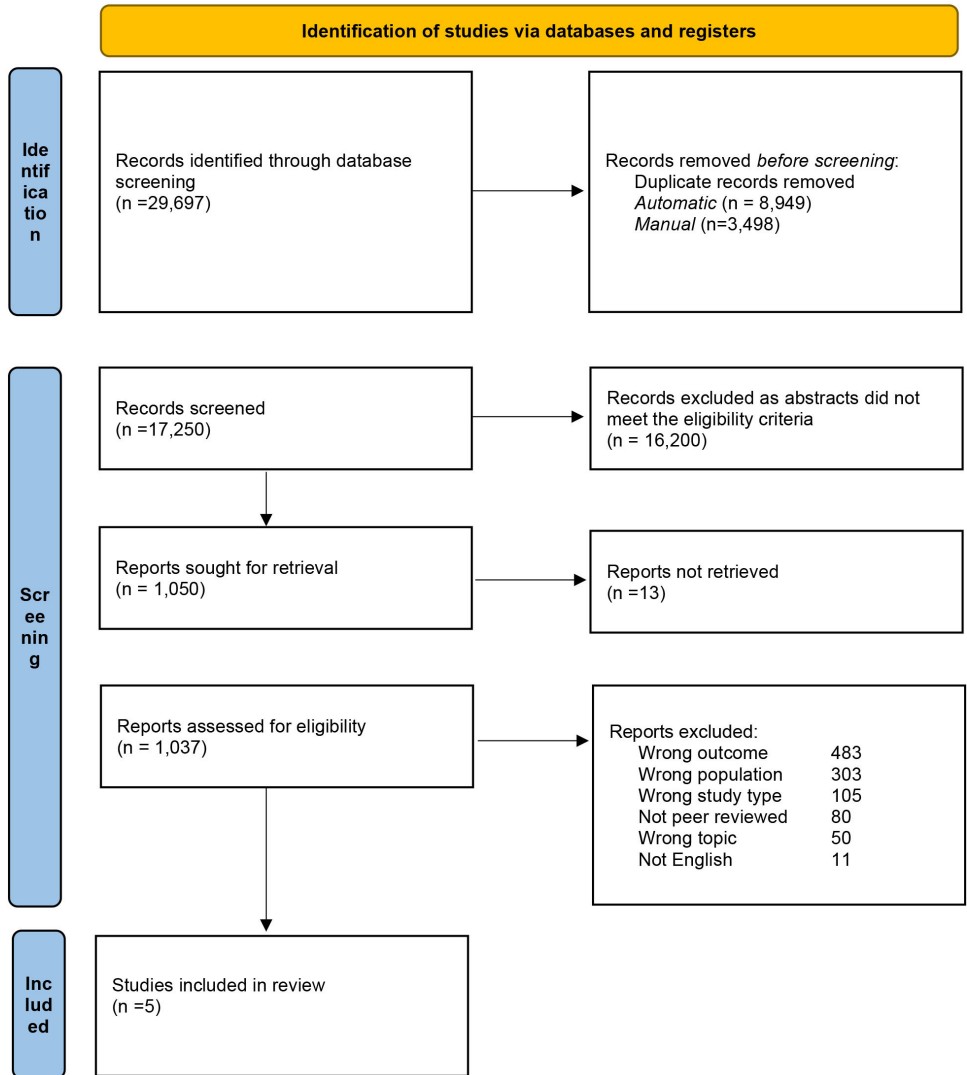

**Figure 1** Preferred Reporting Items for Systematic Reviews and Meta-Analyses flowchart of paper selection process.

as metadata, this was followed by the 'meta summarize' and 'predinterval' commands.

### Patient and public involvement
No patients involved.

### RESULTS
A search in five electronic databases identified 17 250 articles of which 16 200 were excluded based on the eligibility criteria (figure 1, PRISMA diagram). In total, five studies were included in our analysis reporting recovery data of 1011 participants.

Many studies were excluded as they showed a general trend in recovery rather than reporting individual recovery rate. All studies included were longitudinal studies that examined the 1-year outcomes of depressive and/or anxiety symptoms in young people. These papers are summarised in online supplemental table 1.

Several papers did not provide the required information for this review; thus, the authors of each study were contacted to gain the necessary details. There were four

authors that did not reply and therefore those papers were excluded from the analysis.

### Study characteristics
A total of 1011 participants were included in this systematic review. The sample size ranged from 30 to 455 with the age at baseline varying from 12 to 19. Of the five papers in this review four were from high-income countries[16–19] and one from a middle-income country.[20] The majority of the studies reported the 1-year outcome of depression[16–20] with only one study observing the outcomes of anxiety in young people.[18] All young people met either a threshold for elevated symptoms or diagnosis of depression and/or anxiety at baseline, as defined by the authors of the respective studies. All participants in these studies were recruited from education settings.[16–20] The majority of the measures used were self-reporting scales, except for the Kiddie-SADS measure which is clinician led.

### Risk of bias
The quality appraisal of the five studies is presented in online supplemental table 1, with three of the studies

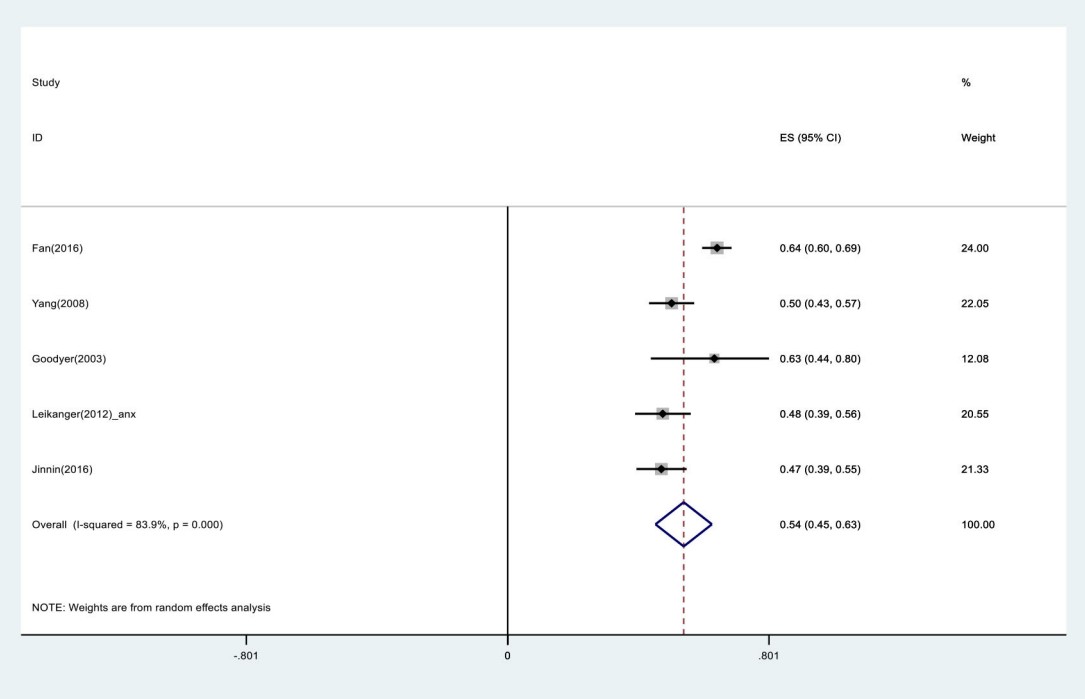

**Figure 2** Forest plot showing pooled proportions and 95% CIs for recovery rate and symptoms reduction of anxiety and/or depression for young people, after 1 year.

assessed to be of good quality scoring (ie, scoring three or four stars in selection, one or two in comparability and two or three stars in outcomes) and only two studies deemed to be of fair reporting quality (ie, scoring two stars in selection, one or two stars in comparability, and two or three stars in outcomes). All studies were deemed representative of the target population and controlled for various confounding variables. The five studies all stated a 1-year follow-up period with little to no follow-up loss. Three studies administered structured interviews to ascertain the exposure; however, none of the studies used a formal assessment to determine the outcome as they relied on the use of self-reported measures (see online supplemental material 4 for full quality appraisal).

### One-year recovery

Five studies provided data for the meta-analysis (see figure 2). The overall pooled proportion of young people to recover (defined as symptom reduction or remission) is 0.54 (0.45 to 0.63), which indicates roughly half (54%) of young people recover without any specific mental health treatment or intervention within a year. Although there is substantial heterogeneity across the eligible studies ($I^2$=84%), to account for the high heterogeneity evident in meta-analyses of observation data, PIs were also calculated (90% PI 0.340 to 0.738), to estimate an interval range in which a future observation will fall.

All five studies reported a recovery rate ranging from 47% to 64%. There was only one study that looked at the recovery of symptoms of anxiety and they found that 48% of young people (95% CI 0.39 to 0.56) had gone from

scoring high on the SCARED scale to low-moderate after a year.[18]

Gender differences for recovery of young people with depression were reported in only one study,[18] which reported that 67 adolescents (47.5% of the high scorers at baseline: 47.8% of girls and 46.4% of boys) had high scores at baseline but low to moderate scores at 12-month follow-up.

### DISCUSSION

The findings of this systematic review and meta-analysis indicate that 54% of young people with symptoms of depression and/or anxiety tend to achieve recovery, defined as symptom reduction below a threshold within a year, when they do not receive specific mental health treatment. The included studies show relatively consistent results with limited variation. Based on the available evidence, the most likely spontaneous prognosis for young people with symptoms of depression and/or anxiety is that about half of them will have recovered after a year. Such process can be understood as a sign of resilience with young people bouncing back from their experience of distress.[21]

There is no systematic review in adults that offers a direct comparison, however literature suggests that minor depression is likely to persist and poses an elevated risk of worsening over 1 year in 60% of older adults.[22]

## Strengths and limitations

This is the first systematic review and meta-analysis to explore 1-year recovery rates of young people with depression and/or anxiety. Symptom reduction across depression and anxiety was shown in similar proportions across the different studies, with markedly small margins which suggest relatively robust findings. Pooled analyses were therefore representative of papers and one study with a larger sample size was not skewing the results. Finally, most articles included were of moderate to high methodological quality and used validated measures with strong psychometric properties. Previous research suggests adolescents are capable of providing valid self-reports of depressive symptoms,[23] strengthening the reliability of our findings.

This review does however have several limitations. First, we have no systematic information about how long symptoms had been present for at baseline. Studies started with cross-sectional identification of participants who met the symptom levels defined as inclusion criteria at the time of recruitment to the cohort study. These studies were likely to miss participants with very short episodes of depression and/or anxiety and recruit participants whose symptoms had already been there for longer. Thus, the spontaneous prognosis of young people experiencing their first symptoms of depression and/or anxiety is likely to be more favourable than the 54% in this review. Second, given the worldwide interest in the mental health of young people, the evidence base with data of a total of 1011 participants is limited, and there was only one study exploring recovery rates of anxiety. Most research studies are conducted in clinical settings where young people are receiving treatment.

It was not possible to access data from the control groups or 'treatment as usual' groups from RCTs, despite requesting this information from authors, furthermore in trials conducted in clinical services it would be unlikely that the control group are not receiving any intervention. Due to the small number of included studies, moderation analysis on patient factors such as age and gender could not be conducted. Third, we focused on only two-time points, that is, baseline and 1-year follow-up, and did not consider what happened in between the two-time points or after the follow-up. For instance, participants who had symptoms after a year might have been symptom-free in between, and participants we regarded as recovered may have relapsed after the follow-up. The findings do therefore not imply that 54% will remain recovered in the future. Although effort was made to include young people not receiving treatment, it is possible that some were accessing support outside of clinical services and the context of this study. Finally, the studies included in this review are highly heterogeneous. Although this suggests that similar recovery rates can be found in very different contexts, it may also challenge whether it is appropriate to derive an overall estimate from that set of studies. The data was also not sufficient to distinguish between narrower age bands within the wide group of young people.

## Implications

Although there is evidence that adolescents with depression go on to experience episodes of depression in later life[24] our review suggests that significant symptom reduction occurs after 1 year in at least half of young people who are affected. Three to nine per cent of teenagers meet the criteria for depression at any one time,[25] and the question arises as to whether they should routinely be considered for specialised treatments or whether one should wait with such decisions for a year by which time about 54% are likely to have recovered without treatment.

A recent systematic review found that although psychotherapies for depression in young people can be effective, more than 60% of those receiving therapy do not respond.[26] Similarly, antidepressants have been found to have only a small therapeutic effect in this group.[27] The decision as to whether specific treatments should be considered is further complicated by the absence of evidence-based predictor variables for (a) who would or would not recover without treatment and (b) who would or would not benefit from specific treatments. School-based and community programmes are gaining traction, especially following the COVID-19 pandemic[28] but further studies are needed to demonstrate long-term outcomes and successful implementation into routine practice.

It is important to analyse and further understand why more than half of young people recover, while other young people continue to report symptoms of depression and anxiety. Further research should identify personal characteristics predicting recovery. Since the predictive value of common sociodemographic characteristics is likely to be limited, such research may have to consider individual experiences, attitudes, family cohesion and community links.

The main task for future research however may be to explore what personal, family and community resources young people can find and use that help them recover from depression and anxiety. Such research may lead to social and public health interventions that can make a difference to large groups of affected young people and substantially improve the recovery rate of 54% found in this review.

Given the relatively small variability of recovery rates in this review, future intervention studies may use the 54% finding of this study as a yardstick to evaluate outcomes, when recruiting and assessing control groups is impractical or impossible.

## CONCLUSION

In conclusion, although it is encouraging to find that half of young people tend to recover without treatment, if almost 50% still have symptoms after 1 year, concerns remain. This is especially so as previous research shows only a small effect of both pharmacological and psychological interventions,

although such treatments through mental health professionals generate significant costs. This calls for further development and evaluation of treatments in mental healthcare, but also interventions delivered within families, schools and communities that prevent mental distress in young people as far as possible and help those experiencing symptoms of depression or anxiety to recover.

**Author affiliations**
[1]Unit of Social and Community Psychiatry (WHO Collaborating Centre for Mental Health Service Development), Queen Mary University of London, London, UK
[2]CRONICAS Center of Excellence in Chronic Diseases, Universidad Peruana Cayetano Heredia, Lima, Peru
[3]Youth Resilience Unit, Centre for Psychiatry and Mental Health, Wolfson Institute of Population Health, Queen Mary University of London, London, UK
[4]Department of Psychiatry and Mental Health, School of Medicine, Universidad de Buenos Aires, Buenos Aires, Argentina

**Contributors** AR and SP came up with the research question. AR, DSS and LH-P led data extraction with KE, FvL, LH, ZJ, FE and JD assisting with title and abstract screening and then full text screening. AR, DSS, KE and VJB analysed the included papers. AR and DSS drafted the manuscript and all provided feedback on the paper before submission. AR is responsible for the overall content, and acts as the guarantor.

**Funding** This work is supported by the Medical Research Council (grant number: MR/S03580X/1).

**Competing interests** None declared.

**Patient and public involvement** Patients and/or the public were not involved in the design, or conduct, or reporting, or dissemination plans of this research.

**Patient consent for publication** Not applicable.

**Ethics approval** Ethical approval was not required for this systematic review and meta-analysis.

**Provenance and peer review** Not commissioned; externally peer reviewed.

**Data availability statement** All data relevant to the study are included in the article or uploaded as supplementary information.

**ORCID iD**
Anna Roach http://orcid.org/0000-0002-0635-0429

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
