## [Reviewer comments · BMJ Open]

ARTICLE DETAILS

TITLE (PROVISIONAL)	One-year recovery rates for young people with depression and/or anxiety not receiving treatment: A systematic review and meta-analysis
AUTHORS	Roach, Anna; Sureshkumar, Diliniya Stanislaus; Elliot, Kathryn; Hidalgo-Padilla, Liliana; van Loggerenberg, Francois; Hounsell, Lauren; Jakaite, Zivile; Esnal, Fernando; Donaghy, Jade; Bird, Victoria; Priebe, Stefan

VERSION 1 – REVIEW

REVIEWER	Garrett, Nick AUT University
REVIEW RETURNED	22-Mar-2023

GENERAL COMMENTS	The article is very well written and easy to read. I only have a couple of small comments - ideally in the data analysis section, there needs to be references for the methodological choices, at present there is only a reference to the STATA software.- In the PRISMA flowchart in figure 1 there are the 16,200 excluded, should state that the abstracts did not meet the eligibility criteria to make it explicitly stated rather than just excluded.
--

REVIEWER	Toenders, Yara Erasmus University Rotterdam
REVIEW RETURNED	24-Mar-2023

GENERAL COMMENTS	This manuscript contains a review of the literature and meta-analysis on the one-year naturalistic course of depression and anxiety in young people. The authors should be applauded for searching five databases. However, after reading the manuscript, there are some remaining issues. The authors state that even though there is a worldwide interest, this evidence base is limited, since only 5 studies were found that study one-year follow-up. In my opinion, drawing conclusions based on only these five studies with considerable heterogeneity is complex. Therefore, I wonder if the authors could broaden their search criteria to also include six-month follow-up studies? Broadening the criteria would lead to the inclusion of many more studies, and would also give the authors the opportunity to compare the results from one-year follow-up studies to six-month follow-up studies. Alternatively, I wonder if the authors could elaborate on why they chose to include only studies with a one-year follow-up period?
--

	Furthermore, some search terms could be added such as 'MDD', 'major depressive disorder', 'naturalistic' and 'wait list' or 'waiting list' The following study might also meet the search criteria: Keller, M. B., & Shapiro, R. W. (1981). Major depressive disorder: Initial results from a one-year prospective naturalistic follow-up study. The Journal of nervous and mental disease, 169(12), 761-768 The authors state that "it was not possible to access data from the control groups or treatment as usual groups from randomized controlled trials". Could the authors explain why this was not possible? Most studies used a 5 different (self-report) measures to measure depressive symptoms, with different cut-offs that were used to determine whether depression was present (or even low, middle, high depression). Could the authors comment on how this might have affected the results? Lastly, waiting for a year can be severely debilitating since only half of the people recover. Could the authors make it more clear from the introduction that the aim is to inform which adolescents could be selected for treatments (instead of waiting for a year)?
--	--

REVIEWER	Mason, Michael University of Tennessee Knoxville, Center for Behavioral Health
REVIEW RETURNED	24-Mar-2023

GENERAL COMMENTS	This manuscript reports on systematic review and meta-analysis of annual recovery rates of young people with depression or anxiety who did not receive treatment. This review addresses an important topic, namely, the spontaneous recovery of young people who experience depressive or anxious symptoms. The study is well described and sufficiently detailed and methods follow established guidelines. This manuscript has the potential to make an important contribution to the literature given current escalation of depression and anxiety among young people. Below are comments, questions, and suggestions that may improve the manuscript.  1. It is unclear how it was known that individuals did not receive treatment during the study period. 2. While the sample of studies is small, the individual sample size seemed large enough to conduct and describe sub-group analyses by age groups. For example, the age range of 12-19 is large enough to be described by young, middle, and late adolescence, or at least young and late adolescence, allowing for understanding some developmental differences. The authors speak to age and gender in the limitation section, but clarification about why individual level data could not be used for these analyses would be helpful 3. Similarly, the severity levels could be used as categorical variables and provide another form of classification by outcome as well as moderating variable interacting with age.
--

VERSION 1 – AUTHOR RESPONSE

Reviewer 1	
- ideally in the data analysis section, there needs to be references for the methodological choices, at present there is only a reference to the STATA software.	We have included references to the methodological choices (e.g. Wilson, E. B. (1927), "Probable Inference, the Law of Succession, and Statistical Inference," Journal of the American Statistical Association, 22, 209-212.)
- In the PRISMA flowchart in figure 1 there are the 16,200 excluded, should state that the abstracts did not meet the eligibility criteria to make it explicitly stated rather than just excluded.	This has been changed to explicitly state that abstracts did not meet eligibility criteria
Reviewer 2	
The authors state that even though there is a worldwide interest, this evidence base is limited, since only 5 studies were found that study one-year follow-up. In my opinion, drawing conclusions based on only these five studies with considerable heterogeneity is complex. Therefore, I wonder if the authors could broaden their search criteria to also include six-month follow-up studies? Broadening the criteria would lead to the inclusion of many more studies, and would also give the authors the opportunity to compare the results from one-year follow-up studies to six-month follow-up studies. Alternatively, I wonder if the authors could elaborate on why they chose to include only studies with a one-year follow-up period? Furthermore, some search terms could be added such as 'MDD', 'major depressive disorder', 'naturalistic' and 'wait list' or 'waiting list'	Episodes of anxiety and depression may be appropriate reactions to life events or adjustments. In traditional diagnostic criteria in psychiatry, these 'appropriate' reactions or adjustments can take up to six months, so that a longer period of time is required to capture outcomes that go beyond that. We decided to focus on one-year follow-ups as a period that is longer than six months but would still be covered in some research. We needed a consistent period of time across studies. We did not include MDD or "major depressive disorder" as a search term as these are terms usually used in clinical services. Additionally we did not include "wait/waiting list" as even in treatment trials, waiting lists are unlikely to be for 1 year
The following study might also meet the search criteria: Keller, M. B., & Shapiro, R. W. (1981). Major depressive disorder: Initial results from a one-year prospective naturalistic follow-up study. The Journal of nervous and mental disease, 169(12), 761-768	This paper would not be included as in the methods section, the authors describe that all 'patients' received treatment in their clinical centre even if the exact type and intensity of treatment varied.
The authors state that "it was not possible to access data from the control groups or treatment as usual groups from randomized controlled trials". Could the authors explain why this was not possible?	We tried to obtain these data from the authors of the studies but did not receive responses. We have added this clarification in the penultimate paragraph of the strengths and limitations section.
Most studies used a 5 different (self-report) measures to measure depressive symptoms, with different cut-offs that were used to determine whether depression was present (or even low, middle, high depression). Could the authors comment on how this might have affected the results?	We followed the design of the studies and needed to assume that the chosen cut-off points were appropriate for the given context. We feel considering how different cut-off points might have influenced the findings would be mere speculation.

Lastly, waiting for a year can be severely debilitating since only half of the people recover. Could the authors make it more clear from the introduction that the aim is to inform which adolescents could be selected for treatments (instead of waiting for a year)?	That was not the aim of the study. As considered in the discussion, professional treatment may be one option for those adolescents, but there are some reasons for caution, because a) services in most parts of the world would not have the capacity to provide treatment for all those adolescents, b) the effectiveness of treatments in this age group is limited, and c) we would never find predictors that are sufficient to identify individual cases with poor prognoses.
Reviewer 3	
It is unclear how it was known that individuals did not receive treatment during the study period.	This is true. We cannot rule out that some participants in one or more of the studies received some type of treatment. However, this was not part of the design of any of the studies and no such treatment was reported. To clarify we have added as a limitation in the final paragraph in the strengths and limitations section.
While the sample of studies is small, the individual sample size seemed large enough to conduct and describe sub-group analyses by age groups. For example, the age range of 12-19 is large enough to be described by young, middle, and late adolescence, or at least young and late adolescence, allowing for understanding some developmental differences. The authors speak to age and gender in the limitation section, but clarification about why individual level data could not be used for these analyses would be helpful	We agree that this would be highly relevant. However, here we can only analyse the variance across studies which is limited with the small number of studies. For analysing the variance within studies we would require merging the full data sets and conduct an individual patient (although the participants here are not patients) data (IPD) meta-analysis. We do not have the full data sets to conduct such an IPD meta-analysis.
Similarly, the severity levels could be used as categorical variables and provide another form of classification by outcome as well as moderating variable interacting with age.	Again, we agree, but this would also require an IPD meta-analysis.

VERSION 2 – REVIEW

REVIEWER	Toenders, Yara Erasmus University Rotterdam
REVIEW RETURNED	06-May-2023
GENERAL COMMENTS	I would like to thank the authors for their reply. I would advise to accept the current manuscript.
REVIEWER	Mason, Michael University of Tennessee Knoxville, Center for Behavioral Health Research
REVIEW RETURNED	24-Apr-2023
GENERAL COMMENTS	The authors have addressed concerns.